# Gender-Related Inequality in Childhood Immunization Coverage: A Cross-Sectional Analysis of DTP3 Coverage and Zero-Dose DTP Prevalence in 52 Countries Using the SWPER Global Index

**DOI:** 10.3390/vaccines10070988

**Published:** 2022-06-21

**Authors:** Nicole E. Johns, Thiago M. Santos, Luisa Arroyave, Bianca O. Cata-Preta, Shirin Heidari, Katherine Kirkby, Jean Munro, Anne Schlotheuber, Andrea Wendt, Kate O’Brien, Anuradha Gupta, Aluísio J. D. Barros, Ahmad Reza Hosseinpoor

**Affiliations:** 1Department of Data and Analytics, World Health Organization, 20 Avenue Appia, 1211 Geneva, Switzerland; johnsn@who.int (N.E.J.); kirkbyk@who.int (K.K.); schlotheuberan@who.int (A.S.); 2International Center for Equity in Health, Federal University of Pelotas, Rua Marechal Deodoro 1160, Pelotas 96020-220, RS, Brazil; tmelo@equidade.org (T.M.S.); larroyave@equidade.org (L.A.); bianca.catapreta@gmail.com (B.O.C.-P.); awendt@equidade.org (A.W.); abarros@equidade.org (A.J.D.B.); 3Postgraduate Program in Epidemiology, Federal University of Pelotas, Rua Marechal Deodoro 1160, Pelotas 96020-220, RS, Brazil; 4Department of Immunization, Vaccines, and Biologicals, World Health Organization, 20 Avenue Appia, 1211 Geneva, Switzerland; heidaris@who.int (S.H.); obrienk@who.int (K.O.); 5Gavi, the Vaccine Alliance, Chemin du Pommier 40, 1218 Geneva, Switzerland; jmunro@gavi.org (J.M.); agupta@gavi.org (A.G.)

**Keywords:** childhood immunization, vaccination, DTP3, zero-dose, gender, gender barriers, empowerment, inequality

## Abstract

Gender-related barriers to immunization are key targets to improve immunization coverage and equity. We used individual-level demographic and health survey data from 52 low- and middle-income countries to examine the relationship between women’s social independence (measured by the Survey-based Women’s emPowERment (SWPER) Global Index) and childhood immunization. The primary outcome was receipt of three doses of the diphtheria-tetanus-pertussis vaccine (DTP3) among children aged 12–35 months; we secondarily examined failure to receive any doses of DTP-containing vaccines. We summarized immunization coverage indicators by social independence tertile and estimated crude and adjusted summary measures of absolute and relative inequality. We conducted all analyses at the country level using individual data; median results across the 52 examined countries are also presented. In crude comparisons, median DTP3 coverage was 12.3 (95% CI 7.9; 16.3) percentage points higher among children of women with the highest social independence compared with children of women with the lowest. Thirty countries (58%) had a difference in coverage between those with the highest and lowest social independence of at least 10 percentage points. In adjusted models, the median coverage was 7.4 (95% CI 5.0; 9.1) percentage points higher among children of women with the highest social independence. Most countries (41, 79%) had statistically significant relative inequality in DTP3 coverage by social independence. The findings suggest that greater social independence for women was associated with better childhood immunization outcomes, adding evidence in support of gender-transformative strategies to reduce childhood immunization inequities.

## 1. Introduction

Immunization is a powerful tool to prevent disease and death, a highly cost-effective public health intervention, and the focus of some of the most successful global health efforts in recent decades. Immunization has contributed significantly to the estimated 59 percent decline in under-five mortality rates since 1990 [1]. Few public health interventions rival immunization in terms of return on investment: every dollar invested in immunization in the world’s lowest income countries yields an estimated direct return of US$20, or US$52 if considering the wider societal impact [2,3]. However, vaccine-preventable disease remains a leading cause of death and disability for children under five worldwide, due in large part to persistent inequalities in immunization coverage [4]. The pandemic has exacerbated these inequalities, with the number of children un- or under-vaccinated with the diphtheria-tetanus-pertussis (DTP) vaccine increasing to over 22 million in 2020 [5]. These children are often concentrated in missed communities that face compounded vulnerabilities and barriers to accessing immunization.

Gender-related barriers, including harmful gender norms, unequal distribution of power and resources, and limited decision-making and mobility for women, are increasingly recognized as persistent obstacles to improving immunization coverage, as well as key correlates of inequality in immunization [6,7]. Addressing gender-related barriers faced by caregivers in seeking vaccine services for their children, by adolescents in accessing vaccine services (e.g., HPV vaccine), by health workers in delivering services, and implementing gender-responsive strategies to improve equitable immunization coverage are key strategic priorities of the Immunization Agenda 2030 (IA2030) [8,9]. Improving gender equity is a means of reducing gender barriers, and is thus recognized both as a key global development goal in itself and as a means to improve the health and well-being of populations more broadly [10].

Despite the recognition of gender equity as a complex construct operating at individual, interpersonal, and structural levels, the examination of gender equity in immunization has previously largely focused on sex differentials in coverage between boys and girls [6,11]. However, the barriers faced by caregivers, specifically mothers, in accessing services for their children are significant determinants of coverage. Gender-related barriers faced by caregivers include inequities in education, mobility, control of resources, and decision-making, harmful gender norms, and gender-based violence [6,11,12]. Several studies have examined the relationship between child immunization coverage and maternal socioeconomic characteristics such as wealth and education [4,13], but less work has documented the association with maternal indicators of gender-related barriers or gender equity, such as empowerment, agency, and autonomy.

The Survey-based Women’s emPowERment (SWPER) Global Index was developed to capture women’s empowerment at the individual level using measures comparable across time and geography [14]. The SWPER Global Index uses Demographic and Health Survey (DHS) program data from up to 63 low- and middle-income countries (LMICs) to construct scales in three domains including the domain analyzed here, social independence. This domain contains items intended to capture pre-conditions which enable women to achieve their goals, and therefore to capture elements which contribute to agency and autonomy: school attainment, access to information, age at key life events, and age difference from partner. Theoretically, a woman with greater social independence may have better direct access to immunization services, and may also be more likely to have better health literacy, freedom of movement, and autonomy to advocate for and act on needs for herself and her children within the household, thereby increasing the chances of immunizing her children [6]. Though mothers are often assumed to be responsible for the medical care of children, including their vaccination (or lack thereof), the decision about whether or not to vaccinate a child may also be influenced by intra-household gender dynamics as well as other gendered socioeconomic factors (such as affordability, distance, and timing of services) [12,15]. A mother’s social independence, as a measure of gender equity more broadly, may thus be relevant to her child’s immunization receipt whether or not she is the one most responsible for her child’s healthcare.

The component items of the SWPER Global Index social independence domain have been shown to be positively correlated with many individual and child health outcomes, including reduced infant and child mortality [16], reproductive, maternal, newborn, and child health interventions [17], and antenatal care, skilled attendance at birth, contraceptive use, nutritional status, reduced exposure to violence, and full vaccination [18]. A 2015 systematic review found that greater household and healthcare decision-making power by women was associated with higher odds of complete childhood immunization [19]. Subsequent analyses have found similar associations in Kenya [20], Ethiopia [21], and the Democratic Republic of the Congo [22].

Few published studies have examined gender-related barriers to care and immunization coverage using individual-level data in nationally representative samples. Only one published study has examined the association between the SWPER Global Index social independence domain and immunization, as measured by zero-dose DTP vaccine prevalence, an indicator of a lack of immunization service utilization [23]. Across 50 countries, greater empowerment as measured by the social independence SWPER domain was associated with lower zero-dose DTP prevalence. We expand on these analyses, examining an additional immunization outcome (DTP3 coverage), and controlling for additional potentially related characteristics. 

In this study, our objective is to assess gender-related inequality in childhood immunization coverage using a measure of women’s agency and autonomy. Specifically, we test the hypothesis that the social independence domain of the SWPER Global Index for a child’s mother will be positively associated with immunization coverage of that child in individual-level analyses across a range of LMIC contexts. 

## 2. Materials and Methods

### 2.1. Data Source

We used child-level data from the DHS program, including all LMICs with recent surveys (2010 onward) containing all necessary items for creating the SWPER Global Index social independence domain and estimating immunization coverage (items detailed below). Where multiple surveys were available for a country, the most recently available survey was used. This resulted in the inclusion of data from 52 countries, with survey years ranging from 2010–2019. Detail on the design and methodology of the DHS have been published elsewhere [24].

The study sample was limited to children aged 12–35 months, born to women married or in a union for whom all SWPER Global Index social independence domain items were available. For most (47) countries, data were available for children aged 12–35 months, although for five countries (Dominican Republic, Egypt, Kyrgyzstan, Peru, and Tajikistan), we studied children aged 18–35 months due to data availability and to make our coverage indicator consistent with results from prior publications and from published DHS survey reports.

### 2.2. Indicators

The primary predictor of interest is women’s social independence, specifically the social independence domain of the SWPER Global Index. The SWPER Global Index uses standard DHS survey data from LMICs; significant detail regarding development and validation of the index has been published elsewhere [14]. The social independence domain measure is based on six items: frequency of reading a newspaper or magazine; years of completed schooling; difference in years of schooling between the woman and her partner; age difference between the woman and her partner; age at first cohabitation; and age at first birth (imputed if nulliparous). Additional detail on the calculation of this measure is published elsewhere [14]. Given that it is a standardized measure, a value of 0 represents the global average social independence among LMICs, while positive values represent greater social independence than the global average and negative values represent lesser social independence than the global average. We present crude findings by globally-defined low, medium, and high tertiles, and otherwise analyzed the SWPER Global Index social independence domain as a continuous measure.

The outcomes of interest included two measures of child immunization coverage. The primary outcome was receipt of three doses of DTP vaccine (DTP3 immunization coverage), which is a common metric of immunization service coverage, indicating full receipt of a vaccine sequence. As a secondary analysis, we also examine the absence of DTP vaccine receipt (zero-dose DTP prevalence), a negative outcome indicating a lack of immunization service utilization. 

### 2.3. Summary Measures of Inequality

We assessed inequality in immunization coverage by women’s social independence via two measures of inequality: the Slope Index of Inequality (SII) and the Concentration Index (CIX); we calculated both measures within each examined country and present overall median results.

The SII is a measure of absolute inequality and is calculated via regression, where the ranked values of the inequality dimension (here, social independence) are scaled to range from 0 to 1. The SII is the slope of the resultant regression line, and here represents the absolute difference in immunization coverage between children of mothers with the highest and lowest values of social independence.

The CIX is a measure of relative inequality whereby individuals are ranked by the dimension of inequality being assessed (in this case, social independence) on the *x*-axis, and the cumulative distribution of the outcome of interest (immunization coverage) on the *y*-axis, with the CIX calculated as twice the area between the x = y (equality) line and the plotted curve, multiplied by 100. CIX values range from −100 (coverage concentrated completely in children of women with lowest social independence) to 100 (coverage concentrated completely in children of women with highest social independence).

For both SII and CIX, a value of 0 indicates no inequality, while positive values indicate greater coverage among children of women with higher social independence, and negative values indicate greater coverage among children of women with lower social independence.

### 2.4. Analyses

We conducted all analyses at the individual level by country; cross-country median results are also presented. First, we present summary information on immunization coverage and the SWPER Global Index social independence score. Next, we conducted unadjusted and adjusted logistic regression models to calculate crude and adjusted SII, respectively, of immunization coverage by women’s social independence. Covariates included in adjusted models of SII were selected a priori based on established associations with immunization coverage and consistent availability across DHS surveys: sex of child, child birth order (continuous), mother’s age (categorized as 15–19, 20–29, 30–39, 40–49), household wealth quintile, household place of residence (urban/rural), and household sub-national region (country-specific). Finally, we calculated the crude CIX measure of immunization coverage by social independence for all countries.

For primary analyses, we used the child age range of 12–35 months rather than the more standard 12–23 months to increase power due to small numbers of zero-dose DTP outcome events. As a sensitivity analysis to minimize potential recall bias and to more closely parallel published figures on immunization coverage indicators, we replicated all analyses limiting the sample to children aged 12–23 months. 

Statistical significance was set at *p* < 0.05 for all comparisons including SII and CIX, and 95% confidence intervals (CIs) are reported throughout. All estimates and respective standard errors accounted for the multi-stage sample design using provided survey structure and weight information. All analyses were conducted using R 4.1.0 and STATA 17. 

### 2.5. Ethical Approval

Ethical approval for DHS data collection was obtained by the national institutions that carried out the surveys, and all analyzed data sets were anonymized and publicly-available.

## 3. Results

### 3.1. Descriptive Statistics

The final analytic sample included 52 countries: 17 from West and Central Africa, 16 from Eastern and Southern Africa, six from East Asia and the Pacific, five from Latin America and the Caribbean, five from South Asia, three from Eastern Europe and Central Asia, and one from North Africa (see Appendix A). These included 24 low income, 22 lower-middle income, and six upper-middle income countries. In total, 187,386 children were analyzed (range 512–16,946 per survey), born to 181,127 mothers (range 498–16,341 per survey). 

Median DTP3 immunization coverage in the 52 countries was 79.3% (95% CI 71.7%; 86.5%), ranging from 31.5% in Chad 2014 to 98.7% in Rwanda 2014 (Table 1). Median zero-dose DTP prevalence was 8.9% (95% CI 5.1%; 14.3%), ranging from 0.6% in Honduras 2011 to 44.7% in Chad 2014 (Appendix A). Though the indicators were highly correlated (Pearson’s r = −0.96), there was wide variation in DTP3 immunization coverage at comparable levels of zero-dose DTP—for example, both South Africa and Zimbabwe had 11% zero-dose DTP prevalence, but DTP3 immunization coverage was 56% in South Africa and 80% in Zimbabwe.

Across the 52 study countries, the mean social independence score was 0 (by scale design), and the median score was −0.07. There was wide variation between countries, with Niger presenting the lowest mean score (−0.87) and Armenia the highest (0.91). 

### 3.2. Overall Coverage by SWPER Global Index Social Independence Domain Tertile

DTP3 immunization coverage by social independence tertile is presented in Figure 1 and Table 1. Across all countries, median DTP3 immunization coverage was 85.0% (95% CI 79.7%; 88.7%) among the highest tertile, 79.2% (95% CI 72.9%; 85.9%) among the middle tertile, and 74.8% (95% CI 64.8%; 79.7%) among the lowest tertile of social independence; this represents a median difference in DTP3 coverage of 10.2% between the highest and lowest tertiles of social independence. In nearly all countries (50/52), DTP3 immunization coverage was higher among the highest tertile of social independence than the lowest (in countries with <10 respondents in the low empowerment tertile, the highest tertile was compared with the middle tertile). In both countries where this was not the case (Armenia & Gambia), coverage rates were within 2% between the highest and lowest tertiles, with overlapping 95% CIs around coverage estimates.

Across all countries, median prevalence of zero-dose DTP was 6.3% (95% CI 3.2%; 8.4%) among the highest tertile, 8.6% (95% CI 4.6%; 15.5%) among the middle tertile, and 12.4% (95% CI 7.0%; 20.8%) among the lowest tertile of social independence; this represents a median difference in zero-dose DTP prevalence of 6.1% between the lowest and highest tertile of social independence (Appendix A).

### 3.3. Absolute Inequality

Across all countries, median crude SII for DTP3 immunization coverage was 12.3 (95% CI 7.9; 16.3), meaning that, after accounting for all children within the country, overall DTP3 immunization coverage was 12.3 percentage points higher among the children of women with the highest social independence score compared with the children of women with the lowest score (Figure 2 and Table 2).

Crude SII for DTP3 immunization coverage was statistically significant in 41 countries (79%), all greater than zero (indicating greater DTP3 immunization coverage among children of women with greater social independence). Crude SII was greater than or equal to 10 in 30 countries (indicating a difference in DTP3 immunization coverage between those with highest and lowest social independence in the country of at least 10 percentage points). 

Nigeria had the highest SII of examined countries (64.8, 95% CI 61.1; 68.5), and Angola, Cameroon, the Democratic Republic of the Congo, Côte d’Ivoire, Ethiopia, Haiti, Myanmar, Pakistan, Papua New Guinea, and the Philippines also had crude SII greater than 20. Adjustment for related characteristics meaningfully attenuated SII findings; median adjusted SII for DTP3 immunization coverage was 7.4 (95% CI 5.0; 9.1). The adjusted SII for DTP3 immunization coverage was statistically significant in 18 countries (35%), of which 11 had an adjusted SII greater than or equal to 10. 

The median crude SII for zero-dose DTP prevalence was −8.3 (95% CI −11.5; −3.9), meaning that, after accounting for all children within the country, overall zero-dose DTP prevalence was 8.3 percentage points lower among the children of women with the highest social independence domain score compared with the children of women with the lowest (Appendix A). Adjustment somewhat attenuated SII findings; median adjusted SII for zero-dose DTP prevalence was −3.5 (95% CI −5.0; −2.2). Adjusted SII for zero-dose DTP prevalence was statistically significant in ten countries (19%), and seven countries had an adjusted SII less than or equal to −10.

### 3.4. Relative Inequality

Across all countries, the median crude CIX for DTP3 immunization coverage was 2.7 (95% CI 1.6; 4.0) (Table 2). Crude CIX for DTP3 immunization coverage was statistically significant in 41 countries (79%), all greater than zero (indicating greater DTP3 immunization coverage among women with greater social independence). CIX was greater than 10 in four countries (Angola, Haiti, Nigeria, and Papua New Guinea). 

Median crude CIX for zero-dose DTP prevalence was −15.5 (95% CI −18.0; −12.3) (Appendix A). Crude CIX for zero-dose DTP prevalence was statistically significant in 40 countries (77%), all less than zero (indicating lower zero-dose DTP prevalence among women with greater social independence). 

### 3.5. Sensitivity Analyses

Restricting the analytic sample to children aged 12–23 months yields largely consistent findings with those from the primary analyses, though estimates generally have greater uncertainty. This is largely due to the reduced sample size, from 187,386 to 96,123. Though prevalence and crude indicators remained remarkably consistent across the age range used, fewer countries had statistically significant inequality across social independence scores when utilizing the narrower age range (see Appendix A for full details). For example, while 27 countries (compared to 30 for age 12–35) had statistically significant crude SII greater than 10 for DTP3 coverage among children age 12–23 months, only four (compared to 11 for age 12–35) had significant SII greater than 10 after adjustment.

## 4. Discussion

Findings from this study of data from 52 LMICs suggest that greater social independence for women was significantly associated with increased DTP3 immunization coverage and decreased zero-dose DTP prevalence within most countries. In nearly all countries, DTP3 coverage was higher among children of women in the highest social independence tertile compared to the lowest. Across the study sample, there was a 12.3 percentage point gap in DTP3 coverage between children of women with the lowest and highest social independence scores. In more than half (30/52) of countries, there was a difference of more than 10 percentage points in DTP3 coverage between those with the highest and lowest social independence scores. In 11 countries, this difference was more than 20 percentage points, with the largest difference observed in Nigeria (65 percentage points). 

Findings for zero-dose prevalence were consistent with DTP3 coverage findings and with prior research [23]. For example, we found an overall decrease in zero-dose prevalence of 8.3 percentage points between children of women with the lowest and highest social independence scores. Zero-dose prevalence among children of women in the highest social independence tertile was nearly half that of children of women in the lowest social independence tertile (6.3% vs. 12.4%). 

Differences in immunization by social independence were somewhat, but not fully, reduced by controlling for other child-, mother-, and household-level characteristics. The attenuation may in part be due to the cross-sectional nature of the analyses and possible bi-directional relationship between social independence and the child, mother and household characteristics. For example, adjusted models accounted for child birth order, and in many countries, higher birth order was significantly associated with lower DTP3 coverage. Lower parity for women (and thus lower birth order for the child) may be the result of greater reproductive agency associated with greater social independence (due to higher education, lesser age gap from partner, later age at first birth, and greater access to information). Lower parity may also enable greater social independence through greater likelihood of education continuation or time and ability to access information. Future research which examines the pathways through which social independence can affect health outcomes for women and their children would help disentangle these associations and inform targets for policy and practice. Though the nature of our analyses do not allow for causal interpretation, our findings add support to previously published studies and calls by major immunization organizations which point to gender-related barriers as a major bottleneck in improving immunization coverage and equity [7]. The SWPER Global Index social independence domain captures a number of items identified as gender-related barriers to immunization: years of education captures low education; access to newspapers and magazines proxies access to information more generally; age and education difference from a partner is often related to power and agency within the home. Yet, other important types of gender-related barriers women face in accessing immunization services for their children are not captured, such as low health literacy, mobility and transportation, and gender-based violence. Furthermore, barriers experienced by caregivers are just one of many interrelated factors that contribute to gender-related barriers to care. Other pertinent barriers include poor or unsafe working conditions, limited training opportunities for women, and a lack of women’s representation in decision-making positions, among others. Strategies to reduce gender-related barriers to immunization should therefore address individual women’s social independence and empowerment, and embody gender transformative approaches, targeting harmful gender norms, structural gender inequities within healthcare, and the broader gender dynamics surrounding health decisions [11]. Multisectoral approaches and partnerships to address gender-related barriers more holistically, while outside of the traditional scope of the immunization sector and likely not the most cost-effective way to improve coverage in the short-term [25], will lead to more sustainable change and will have positive effects beyond the health of individuals. These efforts can include a range of approaches such as gender mainstreaming in relevant stakeholder organizations or collaboration with female-run and female-centered community organizations [26,27]. Such multisectoral approaches align with recent calls from key global stakeholders in immunization, including IA2030 and Gavi 5.0 [8,28].

Findings from this study should be considered in light of several limitations. First, the SWPER Global Index, from which the social independence indicator was drawn, was limited to LMIC contexts with a full DHS, and to mothers who are married or in a union. Findings from other settings and populations may differ. Second, the maternal indicator of social independence captures a number of elements related to gender equality and women’s agency, autonomy, and empowerment. These analyses indicate an association with children’s immunization coverage; however, they do not provide insight into which of these factors are associated, nor the underlying pathways. While the SWPER Global Index is the only multidimensional and individual-level indicator of women’s empowerment available that is comparable across time and countries, it does not fully capture the construct of empowerment. Due to this and the cross-sectional nature of the data, these analyses cannot determine a causal relationship between women’s social independence, or women’s empowerment more generally, and immunization coverage for their children. The significance and consistency of our findings, however, add support to theorized causal relationships [11]. Third, there was a very small number of children with zero-dose DTP in some countries, limiting the power to detect significant differences by social independence score. Findings for zero-dose DTP should be interpreted with caution.

Findings from this study can inform and would be bolstered by additional future research. The analyses presented here were conducted at the individual level within countries and therefore do not examine national-level characteristics which may modify or mediate the relationship between social independence and immunization coverage. Ecological analyses using national-level data would enable cross-national comparison and could bolster the comparability of findings across country contexts. Additionally, this study focused on gender-related barriers to immunization coverage, and did not consider the relative influence of a range of potential barriers to care. Future research which considers gender-related barriers to immunization in tandem with others (such as geographic location, vaccine stockouts, inadequate staffing, and vaccine hesitancy, among others) would allow for an understanding of the relative importance of these factors, and could further inform practical guidance on the most important and cost-effective methods to improve coverage. Finally, this study examined only two outcomes with the DTP vaccine; the relationship between social independence and coverage may differ for vaccines delivered on different schedules or with alternate delivery mechanisms. Future research which examines alternate vaccines would strengthen the findings presented here.

## 5. Conclusions

Our study across 52 LMICs suggests that greater social independence for women was associated with better childhood immunization outcomes within most countries, namely higher DTP3 coverage and lower zero-dose DTP prevalence. Broadly, these findings support prior research and calls to action which identify gender-related factors as key barriers to childhood immunization coverage [6,8] and add further evidence in support of gender-transformative strategies to reduce childhood immunization inequities. 

## Figures and Tables

**Figure 1 vaccines-10-00988-f001:**
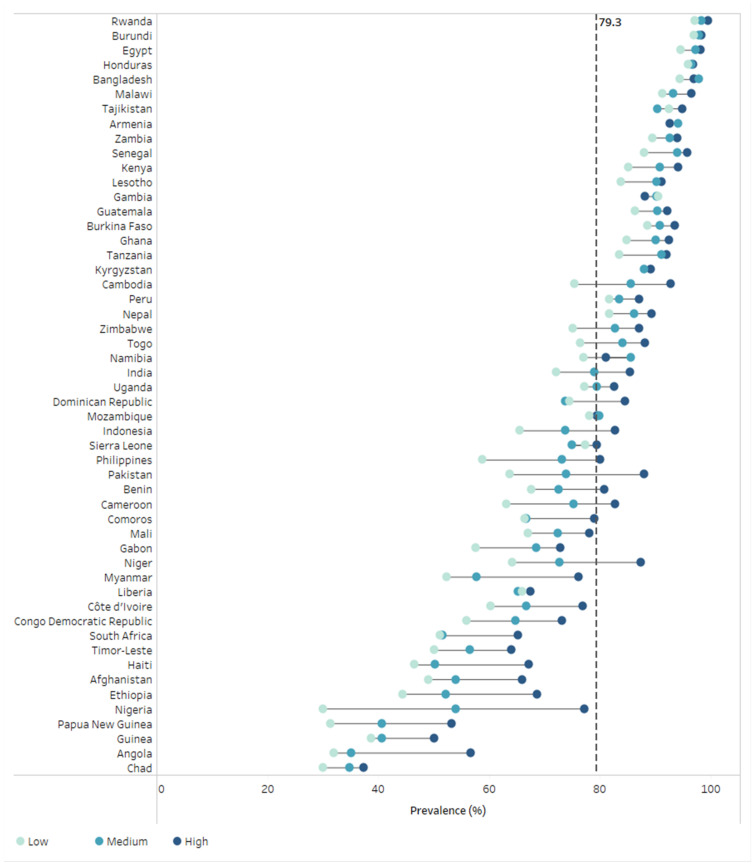
DTP3 immunization coverage by social independence tertile in 52 countries (DHS 2010–2019). *Colored circles indicate social independence tertiles within each country. The black vertical dashed line indicates the median national coverage across 52 countries. DHS = Demographic and Health Surveys. DTP = diphtheria-tetanus-pertussis*.

**Figure 2 vaccines-10-00988-f002:**
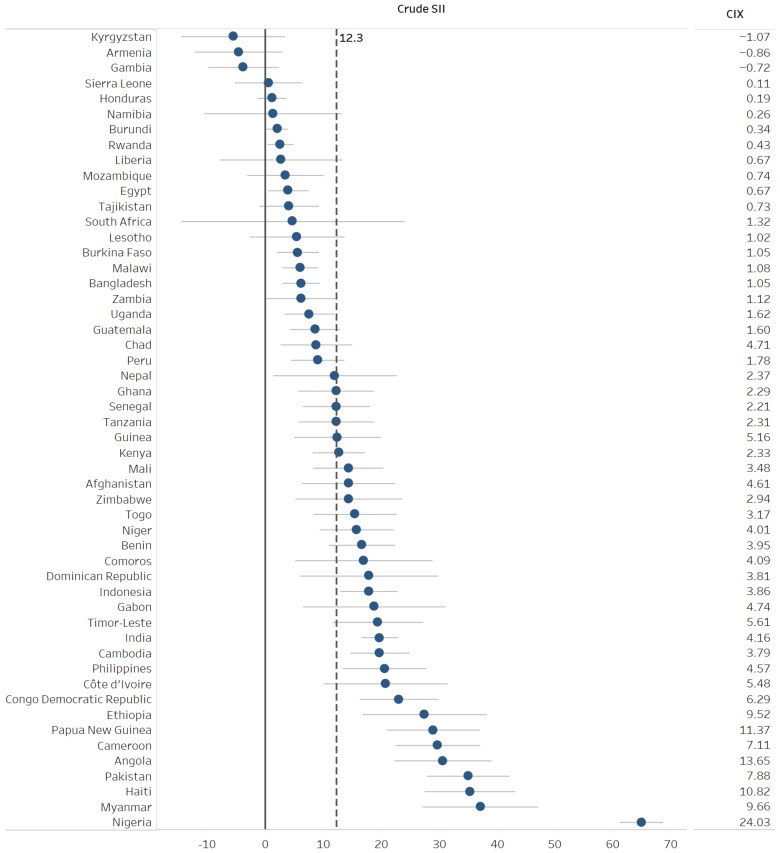
Inequality in DTP3 immunization coverage in women’s social independence in 52 countries: Crude Slope Index of Inequality and Concentration Index (DHS 2010–2019). *Blue circles show the crude SII for each country. Horizontal grey lines indicate 95% confidence intervals for crude SII estimates. The black vertical dashed line indicates the median crude SII value across 52 countries. The black vertical solid line indicates the SII value of no inequality (zero). CIX = Concentration Index. DHS = Demographic and Health Surveys. DTP = diphtheria-tetanus-pertussis. SII = Slope Index of Inequality*.

**Table 1 vaccines-10-00988-t001:** DTP3 coverage overall and by SWPER Global Index social independence domain tertile, children age 12–35 months in 52 LMICs.

Country	National Average (%)			Low Social Independence Tertile (%)			Medium Social Independence Tertile (%)			High Social Independence Tertile (%)		
	Prevalence	LL	UL	Prevalence	LL	UL	Prevalence	LL	UL	Prevalence	LL	UL
** *Overall median* **	*79.3*	*71.7*	*86.5*	*74.8*	*64.8*	*79.7*	*79.2*	*72.9*	*85.9*	*85.0*	*79.7*	*88.7*
Afghanistan	52.9	49.4	56.5	49.0	45.3	52.8	53.9	49.7	58.1	66.0	58.2	73.0
Angola	38.6	35.7	41.6	32.0	28.5	35.8	35.1	31.6	38.8	56.7	50.6	62.6
Armenia	93.0	90.4	94.9	-	-	-	94.0	88.9	96.9	92.6	89.6	94.7
Bangladesh	96.1	95.0	97.0	94.4	92.5	95.8	97.8	96.7	98.5	96.9	94.7	98.3
Benin	71.8	69.5	74.0	67.5	64.2	70.6	72.6	69.7	75.3	80.8	77.2	83.9
Burkina Faso	89.5	88.0	90.9	88.5	86.5	90.2	90.9	88.7	92.6	93.5	89.7	95.9
Burundi	97.7	97.1	98.1	96.9	95.6	97.9	97.8	97.1	98.4	98.3	97.2	98.9
Cambodia	87.1	85.0	88.9	75.3	70.1	79.9	85.6	82.6	88.1	92.7	90.5	94.4
Cameroon	71.7	68.5	74.6	63.1	58.4	67.5	75.2	71.0	79.0	82.7	78.8	86.0
Chad	31.5	29.3	33.7	30.0	27.6	32.4	34.8	31.1	38.6	37.4	30.4	44.9
Comoros	70.6	65.9	74.9	66.5	59.5	72.8	66.8	58.0	74.5	78.9	72.5	84.2
Côte d’Ivoire	64.5	60.2	68.6	60.2	54.6	65.6	66.7	61.2	71.7	76.8	68.8	83.2
Democratic Republic of the Congo	62.7	59.6	65.6	56.0	51.9	60.0	64.7	60.9	68.4	73.1	68.6	77.2
Dominican Republic	79.3	75.1	83.0	74.5	63.5	83.0	73.7	65.9	80.2	84.5	79.1	88.8
Egypt	97.3	96.4	98.0	94.6	89.9	97.2	97.3	96.1	98.1	98.1	97.2	98.8
Ethiopia	49.6	45.8	53.3	44.4	40.1	48.7	52.2	46.3	58.0	68.7	60.0	76.2
Gabon	67.5	63.4	71.5	57.5	49.2	65.5	68.5	61.7	74.6	72.8	65.4	79.1
Gambia	89.9	88.4	91.3	90.5	88.2	92.4	90.2	87.5	92.3	88.0	83.3	91.6
Ghana	89.2	86.6	91.4	84.8	79.8	88.7	90.0	86.7	92.6	92.4	89.7	94.5
Guatemala	89.9	88.5	91.1	86.3	83.6	88.7	90.4	88.4	92.1	92.1	90.0	93.8
Guinea	40.9	37.5	44.3	38.8	34.7	43.0	40.6	35.7	45.7	50.1	44.4	55.8
Haiti	56.3	52.7	59.9	46.4	40.9	52.0	50.2	45.1	55.4	67.2	62.5	71.5
Honduras	96.4	95.6	97.1	95.9	94.3	97.1	96.5	95.1	97.5	96.8	95.1	98.0
India	80.3	79.4	81.2	72.0	70.0	74.0	79.0	77.5	80.4	85.5	84.1	86.7
Indonesia	78.5	76.9	80.0	65.4	60.1	70.4	73.7	70.7	76.5	82.7	81.3	84.1
Kenya	90.5	89.1	91.8	85.1	81.9	87.8	90.8	88.9	92.5	94.1	90.9	96.2
Kyrgyzstan	88.9	86.5	90.9	-	-	-	88.0	82.7	91.8	89.2	86.5	91.4
Lesotho	90.0	87.6	92.0	83.8	67.6	92.7	90.2	86.5	92.9	91.1	86.5	94.2
Liberia	66.0	61.7	70.0	66.0	59.9	71.6	65.2	59.2	70.8	67.4	57.8	75.8
Malawi	93.0	92.1	93.9	91.3	89.5	92.8	93.1	91.8	94.3	96.4	94.8	97.6
Mali	69.9	66.8	72.8	67.1	63.3	70.6	72.4	68.4	76.0	78.0	72.5	82.7
Mozambique	78.9	76.3	81.3	78.1	74.6	81.2	79.8	76.4	82.8	79.5	74.5	83.7
Myanmar	66.4	62.5	70.0	52.4	44.1	60.5	57.7	51.1	64.0	76.2	72.0	79.9
Namibia	81.7	78.0	84.8	77.0	66.6	84.9	85.6	79.5	90.1	81.1	76.3	85.1
Nepal	85.3	82.4	87.8	81.7	76.9	85.7	86.1	82.5	89.0	89.3	82.9	93.5
Niger	66.7	63.7	69.6	64.1	61.0	67.2	72.7	68.2	76.7	87.4	80.5	92.1
Nigeria	49.1	47.2	50.9	30.1	27.9	32.3	53.9	51.4	56.4	77.2	75.1	79.2
Pakistan	75.9	72.4	79.1	63.7	58.6	68.5	73.9	68.6	78.6	88.0	85.0	90.5
Papua New Guinea	43.7	40.1	47.3	31.3	25.8	37.4	40.6	36.3	45.1	53.3	48.7	57.8
Peru	85.6	84.1	86.9	81.6	76.9	85.6	83.4	80.7	85.8	87.0	85.2	88.6
Philippines	76.5	74.3	78.5	58.7	50.7	66.3	73.2	69.0	76.9	80.1	77.7	82.2
Rwanda	98.7	97.9	99.2	97.1	93.8	98.7	98.3	96.6	99.1	99.5	98.9	99.8
Senegal	91.8	89.5	93.6	88.0	84.1	91.0	93.9	91.7	95.5	95.7	92.3	97.7
Sierra Leone	76.9	74.7	79.0	77.4	74.6	80.0	74.9	71.2	78.2	79.5	75.0	83.3
South Africa	61.4	55.6	66.9	51.2	25.4	76.4	51.5	39.9	63.0	65.2	58.7	71.2
Tajikistan	93.0	90.9	94.7	92.5	83.8	96.7	90.4	86.1	93.5	94.8	92.8	96.2
Timor-Leste	59.1	56.1	61.9	50.0	44.3	55.8	56.5	52.1	60.7	64.0	60.5	67.4
Togo	82.3	79.4	84.9	76.4	71.7	80.6	84.1	80.7	86.9	88.1	84.2	91.1
Uganda	79.4	77.8	80.9	77.1	74.7	79.4	79.4	76.8	81.9	82.6	79.8	85.0
United Republic of Tanzania	89.2	86.8	91.2	83.5	78.4	87.6	91.1	88.8	92.9	92.0	89.7	93.8
Zambia	91.9	90.1	93.3	89.4	86.0	92.0	92.6	90.5	94.2	94.0	89.1	96.7
Zimbabwe	83.0	80.1	85.4	75.1	67.6	81.4	82.7	78.7	86.0	87.1	83.3	90.1

LL: Lower limit 95% confidence interval; LMIC: Low- and middle-income countries; UL: Upper limit 95% confidence interval.

**Table 2 vaccines-10-00988-t002:** Measures of inequality in DTP3 coverage by SWPER Global Index social independence domain, children age 12–35 months in 52 LMICs.

Country	Crude SII (Percentage Points)	LL (Percentage Points)	UL (Percentage Points)	Adjusted SII (Percentage Points)	LL (Percentage Points)	UL (Percentage Points)	CIX	LL	UL
** *Overall median* **	*12.3*	*7.9*	*16.3*	*7.4*	*5.0*	*9.1*	*2.7*	*1.6*	*4.0*
Afghanistan	14.3	6.4	22.2	9.8	2.7	16.9	4.6	2.1	7.1
Angola	30.6	22.3	38.9	13.9	2.8	25.0	13.7	9.9	17.4
Armenia	−4.7	−12.2	2.8	−0.4	−2.5	1.7	−0.9	−2.2	0.5
Bangladesh	6.1	2.9	9.3	6.3	−0.5	13.1	1.1	0.5	1.6
Benin	16.6	11.0	22.3	7.3	−0.4	15.0	3.9	2.5	5.4
Burkina Faso	5.5	1.9	9.1	0.4	−2.1	3.0	1.0	0.4	1.7
Burundi	2.0	0.2	3.7	3.5	−2.2	9.2	0.3	0.0	0.6
Cambodia	19.7	14.6	24.7	12.0	1.4	22.7	3.8	2.8	4.8
Cameroon	29.7	22.5	36.9	11.0	−1.6	23.7	7.1	5.2	9.0
Chad	8.7	2.6	14.8	2.4	−1.5	6.3	4.7	1.4	8.0
Comoros	16.9	5.1	28.7	5.7	−13.6	24.9	4.1	1.2	7.0
Côte d’Ivoire	20.7	10.0	31.3	11.8	−2.0	25.5	5.5	2.5	8.5
Democratic Republic of the Congo	23.0	16.3	29.8	10.5	2.4	18.5	6.3	4.3	8.3
Dominican Republic	17.8	5.9	29.6	17.1	−0.6	34.8	3.8	1.2	6.4
Egypt	3.9	0.5	7.3	5.0	−2.1	12.1	0.7	0.1	1.2
Ethiopia	27.5	16.8	38.1	9.2	−1.5	19.9	9.5	5.7	13.3
Gabon	18.7	6.5	31.0	19.9	1.7	38.1	4.7	1.6	7.8
Gambia	−3.8	−9.9	2.2	1.8	−8.3	12.0	−0.7	−1.9	0.4
Ghana	12.1	5.6	18.6	15.8	1.9	29.8	2.3	1.0	3.5
Guatemala	8.5	4.2	12.8	2.5	−5.2	10.2	1.6	0.8	2.4
Guinea	12.4	5.0	19.8	6.7	0.0	13.4	5.2	2.0	8.3
Haiti	35.2	27.5	43.0	9.7	−1.9	21.3	10.8	8.2	13.5
Honduras	1.1	−1.4	3.6	−7.0	−18.4	4.4	0.2	−0.3	0.6
India	19.6	16.6	22.7	9.6	1.0	18.2	4.2	3.5	4.8
Indonesia	17.8	12.9	22.7	14.9	6.6	23.2	3.9	2.8	5.0
Kenya	12.6	8.1	17.1	6.9	−1.6	15.3	2.3	1.5	3.1
Kyrgyzstan	−5.6	−14.5	3.3	0.0	−1.3	1.3	−1.1	−2.8	0.6
Lesotho	5.4	−2.7	13.5	7.5	−15.0	29.9	1.0	−0.5	2.6
Liberia	2.6	−7.9	13.1	−0.1	−15.9	15.6	0.7	−2.0	3.4
Malawi	5.9	2.9	9.0	11.3	3.9	18.7	1.1	0.5	1.6
Mali	14.3	8.2	20.3	8.1	−0.3	16.5	3.5	2.0	5.0
Mozambique	3.4	−3.1	10.0	3.4	−3.6	10.3	0.7	−0.7	2.2
Myanmar	37.1	27.1	47.0	27.4	12.9	41.8	9.7	6.8	12.5
Namibia	1.2	−10.6	13.0	2.9	−9.4	15.3	0.3	−2.2	2.7
Nepal	11.9	1.3	22.5	1.5	−2.4	5.4	2.4	0.3	4.5
Niger	15.7	9.4	22.0	11.8	3.5	20.1	4.0	2.3	5.7
Nigeria	64.8	61.1	68.5	27.6	20.6	34.6	24.0	22.1	25.9
Pakistan	34.9	27.9	42.0	7.6	−3.7	18.8	7.9	6.0	9.7
Papua New Guinea	28.9	20.9	36.9	7.0	0.1	14.0	11.4	8.0	14.8
Peru	9.0	4.4	13.5	8.5	1.2	15.9	1.8	0.9	2.7
Philippines	20.5	13.3	27.7	7.7	−1.9	17.3	4.6	2.9	6.2
Rwanda	2.6	0.4	4.7	0.7	0.0	1.4	0.4	0.1	0.8
Senegal	12.2	6.4	17.9	8.0	−5.8	21.9	2.2	1.2	3.3
Sierra Leone	0.5	−5.3	6.3	1.5	−6.0	9.0	0.1	−1.2	1.4
South Africa	4.7	−14.6	23.9	14.1	−5.3	33.6	1.3	−4.0	6.7
Tajikistan	4.0	−1.1	9.1	5.0	−4.2	14.2	0.7	−0.2	1.7
Timor-Leste	19.4	11.6	27.2	17.2	7.2	27.2	5.6	3.3	7.9
Togo	15.4	8.2	22.5	8.9	0.8	17.0	3.2	1.7	4.7
Uganda	7.5	3.3	11.8	4.8	0.7	8.9	1.6	0.7	2.5
United Republic of Tanzania	12.2	5.8	18.7	2.1	−2.3	6.4	2.3	1.1	3.6
Zambia	6.1	0.0	12.2	2.3	−5.8	10.4	1.1	0.0	2.2
Zimbabwe	14.3	5.2	23.5	0.5	−5.4	6.4	2.9	1.0	4.8

CIX: Concentration index; LL: Lower limit 95% confidence interval; LMIC: Low- and middle-income countries; SII: Slope index of inequality; UL: Upper limit 95% confidence interval.

## Data Availability

All analyses were carried out using publicly available datasets that can be obtained directly from the DHS (dhsprogram.com) and the MICS (mics.unicef.org) websites. Datasets are continuously sourced and updated by the International Center for Equity in Health (equidade.org) as they are released. Analyses used the latest available dataset versions as of 8 December 2021.

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
