# Peer review of "Gender-Related Inequality in Childhood Immunization Coverage: A Cross-Sectional Analysis of DTP3 Coverage and Zero-Dose DTP Prevalence in 52 Countries Using the SWPER Global Index"

_vaccines, 2022, doi:10.3390/vaccines10070988_

Round 1

Reviewer 1 Report

This is an interesting idea that re-iterates the underlying socio-cultural determinants of vaccination.

The results are self-explanatory, suggesting that better gender policies lead to better vaccination coverage. This is probably a proxy for the HDI or a similar development index. It would be great to have a multivariable model to assess which of those carries more variance.

The second problem is data synthesis – you worked with derived data, which will cause a reduction in the data variance, but this means that your results are even better if they manage to withstand this. There is much averaging and assuming sufficient data quality across countries (honestly, I doubt that such data is directly comparable across countries).

Lastly, the problem is an ecological bias, as you did not use the individual data. This is a bit of a problem, as it can reduce the causality. Are there studies that validated the SWPER, SII or CIX survey in the countries included in the analysis? Is there a systematic risk of bias in the direct pooling of these results? I am not sure that the gender-based efforts ts are a cost-sensitive approach to increasing the vaccination. This is simply a reflection, just like I stated before, not necessarily the cause. Without comparing both in the same model, you are not able to say which is causative and, therefore, which is amenable to intervention.

References 26 is misformated.

Did you consider showing the results against the population size/country? Maybe this can give an interesting angle of the problem.

Author Response

Thank you for your review and feedback. Please see attached word file for full response.

Reviewer 2 Report

This is a well-written and valuable contribution to the literature on childhood immunization coverage. 

I would have liked to see the authors cover--at least briefly--the issue of women as the family member responsible for their children's immunizations.  It seems like there is an underlying assumption that getting children immunized is the province of mothers--cross-culturally--and if so, this should be discussed.

It is not a major point, but given that the article's title promises a discussion of "gender-related inequality", I think that examination of the literature about maternal responsibilities in this arena would be a good addition.

Author Response

This is a well-written and valuable contribution to the literature on childhood immunization coverage. 

-We thank Reviewer 2 for their review and feedback.

I would have liked to see the authors cover--at least briefly--the issue of women as the family member responsible for their children's immunizations.  It seems like there is an underlying assumption that getting children immunized is the province of mothers--cross-culturally--and if so, this should be discussed. It is not a major point, but given that the article's title promises a discussion of "gender-related inequality", I think that examination of the literature about maternal responsibilities in this arena would be a good addition.

-This is indeed a relevant area of research; we have added the following brief mention to the introduction to further contextualize the relevance of maternal empowerment, regardless of primary caretaker status: “Though mothers are often assumed to be responsible for the medical care of children, including their vaccination (or lack thereof), the decision about whether or not to vaccinate a child may also be influenced by intra-household gender dynamics as well as other gendered socioeconomic factors (such as affordability, distance, and timing of services). A mother’s social independence, as a measure of gender equity more broadly, may thus be relevant to her child’s immunization receipt whether or not she is the one most responsible for her child’s healthcare.” (Introduction lines 89-95).

Reviewer 3 Report

Thank you for the opportunity to review this interesting paper, which investigates the correlation between female social independence and vaccination in the pediatric population.

I really appreciated the originality of the investigation carried out by the authors. The paper is well conducted and well presented:

the abstract, although not formally structured, follows a correct logic in describing the work.

The Introduction is informative;

the Methods are clearly described and the Results well expounded - I really appreciated the tables and figures constructed;

the Discussion is well argued, but lacks the final paragraph of Conclusions, which I would ask the authors to add.

The references are adequate. There are fewer language typos to check and please avoid starting a new sentence with a number (e.g. lines 244, 246).

Author Response

Thank you for the opportunity to review this interesting paper, which investigates the correlation between female social independence and vaccination in the pediatric population.

-We thank Reviewer 3 for their review and feedback.

I really appreciated the originality of the investigation carried out by the authors. The paper is well conducted and well presented: the abstract, although not formally structured, follows a correct logic in describing the work. The Introduction is informative; the Methods are clearly described and the Results well expounded - I really appreciated the tables and figures constructed;

the Discussion is well argued, but lacks the final paragraph of Conclusions, which I would ask the authors to add.

-Thank you; we have added a ‘Conclusion’ subsection to the end of the Discussion. “4.1 Conclusion. Our study across 52 LMICs suggests that greater social independence for women was associated with better childhood immunization outcomes within most countries, namely higher DTP3 coverage and lower zero-dose DTP prevalence. Broadly, these findings support prior research and calls to action which identify gender-related factors as key barriers to childhood immunization coverage; and add further evidence in support of gender-transformative strategies to reduce childhood immunization inequities.” (Discussion lines 408-415).

The references are adequate. There are fewer language typos to check and please avoid starting a new sentence with a number (e.g. lines 244, 246).

-We have revised sentences which start with numbers (Results lines 271, 274, 282, 292, 297, 300, 303) and have performed an additional copy-edit of the paper.

Reviewer 4 Report

In this study the authors assessed gender-related inequality in childhood immunization coverage. The study included 52 countries from low and middle income to find the relation between the women’s social independence and child immunization. The authors found that median DTP3 coverage was 12.3 and  percentage points was higher among children  of women with the highest social independence compared with children of women with the lowest, and most countries had statistically significant relative inequality in DTP3 coverage by  social independence.

I have some questions

1- The authors focused mainly on 1 vaccine (DTP), more child vaccines should be included in the study such as HBV, mumps, measles, rota, other vaccines given to child.

2- Including countries from high income as control could be benefit.

3-Please include p values between different countries

4-Please compare between countries from the same continent, and find if there are relations/ common factors present in the countries of the same continent.

Author Response

(The authors gave the same response as above.)

Round 2

Reviewer 4 Report

No further comments

Author Response

We thank reviewer 4 for their additional review.